# Circulating Amino Acids and Risk of Peripheral Artery Disease in the PREDIMED Trial

**DOI:** 10.3390/ijms24010270

**Published:** 2022-12-23

**Authors:** Cristina Razquin, Miguel Ruiz-Canela, Estefania Toledo, Clary B. Clish, Marta Guasch-Ferré, Jesús F. García-Gavilán, Clemens Wittenbecher, Angel Alonso-Gómez, Montse Fitó, Liming Liang, Dolores Corella, Enrique Gómez-Gracia, Ramon Estruch, Miquel Fiol, Jose M. Santos-Lozano, Luis Serra-Majem, Emilio Ros, Fernando Aros, Jordi Salas-Salvadó, Frank B. Hu, Miguel A. Martínez-González

**Affiliations:** 1Department of Preventive Medicine and Public Health, IdiSNA (Instituto de Investigación Sanitaria de Navarra), University of Navarra, 31008 Pamplona, Spain; 2CIBER Fisiopatología de la Obesidad y Nutrición (CIBERObn), Instituto de Salud Carlos III, 28029 Madrid, Spain; 3Broad Institute and MIT, Harvard University, Cambridge, MA 02142, USA; 4Department of Nutrition, Harvard T.H. Chan School of Public Health, Boston, MA 02115, USA; 5Department of Public Health, Section of Epidemiology, University of Copenhagen, 1165 Copenhagen, Denmark; 6Novo Nordisk Foundation Center for Basic Metabolic Research, University of Copenhagen, 1353 Copenhagen, Denmark; 7Unitat de Nutrició Humana, Departament de Bioquímica i Biotecnologia, Universitat Rovira i Virgili, 43201 Reus, Spain; 8Institut d’Investigació Sanitària Pere Virgili (IISPV), Hospital Universitari San Joan de Reus, 43204 Reus, Spain; 9SciLifeLab, Food and Nutrition Science Division, Department of Biology and Biological Engineering, Chalmers University of Technology, SE-412 96 Gothenburg, Sweden; 10Bioaraba Health Research Institute, Osakidetza Basque Health Service, Araba University Hospital, University of the Basque Country UPV/EHU, 01009 Vitoria-Gasteiz, Spain; 11Unit of Cardiovascular Risk and Nutrition, Institut Hospital del Mar d’Investigacions Mèdiques (IMIM), 08003 Barcelona, Spain; 12Department of Epidemiology and Biostatistics, Harvard T.H. Chan School of Public Health, Boston, MA 02115, USA; 13Department of Preventive Medicine, University of Valencia, 46100 Valencia, Spain; 14Department of Preventive Medicine, University of Málaga, 29071 Málaga, Spain; 15Department of Internal Medicine, Institut d’Investigacions Biomèdiques August Pi Sunyer (IDIBAPS), Hospital Clinic, University of Barcelona, 08036 Barcelona, Spain; 16Plataforma de Ensayos Clínicos, Instituto de Investigación Sanitaria Illes Balears (IdISBa), Hospital Universitario Son Espases, 07120 Palma de Mallorca, Spain; 17Department of Family Medicine, Research Unit, Distrito Sanitario Atención Primaria Sevilla, 41013 Sevilla, Spain; 18Nutrition Research Group, Research Institute of Biomedical and Health Sciences (IUIBS), University of Las Palmas de Gran Canaria, 35016 Las Palmas de Gran Canaria, Spain; 19Lipid Clinic, Department of Endocrinology and Nutrition, Agust Pi i Sunyer Biomedical Research Institute (IDIBAPS), Hospital Clinic, University of Barcelona, 08036 Barcelona, Spain; 20Channing Division of Network Medicine, Department of Medicine, Brigham and Women’s Hospital, Harvard Medical School, Boston, MA 02115, USA

**Keywords:** peripheral artery disease, amino acids, metabolomics, PREDIMED, case-cohort

## Abstract

Effective prevention and risk prediction are important for peripheral artery disease (PAD) due to its poor prognosis and the huge disease burden it produces. Circulating amino acids (AA) and their metabolites may serve as biomarkers of PAD risk, but they have been scarcely investigated. The objective was to prospectively analyze the associations of baseline levels of plasma AA (and their pathways) with subsequent risk of PAD and the potential effect modification by a nutritional intervention with the Mediterranean diet (MedDiet). A matched case-control study was nested in the PREDIMED trial, in which participants were randomized to three arms: MedDiet with tree nut supplementation group, MedDiet with extra-virgin olive oil (EVOO) supplementation group or control group (low-fat diet). One hundred and sixty-seven PAD cases were matched with 250 controls. Plasma AA was measured with liquid chromatography/mass spectrometry at the Broad Institute. Baseline tryptophan, serine and threonine were inversely associated with PAD (OR_for 1 SD increase_ = 0.78 (0.61–0.99); 0.67 (0.51–0.86) and 0.75 (0.59–0.95), respectively) in a multivariable-adjusted conditional logistic regression model. The kynurenine/tryptophan ratio was directly associated with PAD (OR_for 1 SD increase_ = 1.50 (1.14–1.98)). The nutritional intervention with the MedDiet+nuts modified the association between threonine and PAD (*p*-value interaction = 0.018) compared with the control group. However, subjects allocated to the MedDiet+EVOO group were protected against PAD independently of baseline threonine. Plasma tryptophan, kynurenine/tryptophan ratio, serine and threonine might serve as early biomarkers of future PAD in subjects at a high risk of cardiovascular disease. The MedDiet supplemented with EVOO exerted a protective effect, regardless of baseline levels of threonine.

## 1. Introduction

Peripheral artery disease (PAD) is defined as the occlusion of the arteries that supply the lower extremities, and it is usually caused by atherosclerosis and associated thrombosis [1]. PAD is one of the main causes of atherosclerotic cardiovascular morbidity. The global burden of PAD has increased over the last decades [2,3].

Patients with PAD have a considerably increased risk of major cardiovascular and cerebrovascular events, together with functional impairment. In fact, PAD has been associated with heart failure (HF), atrial fibrillation (AF), myocardial infarction and stroke [4,5,6,7]. Consequently, patients with PAD show a poorer quality of life and elevated mortality rates [2,4,6,8,9,10].

Although symptoms of PAD may appear, such as intermittent claudication or atypical leg symptoms [11], PAD is still commonly underdiagnosed in primary care, with a consequent increase in the aforementioned risks of other hard cardiovascular events and premature death [7]. Its huge disease burden and the substantial proportion of underdiagnosed PAD cases highlight the need to improve early predictions of PAD beyond the use of traditional risk factors [12]. The discovery of new biomarkers of PAD may contribute to designing more effective preventive strategies.

Amino acids (AA) are the building blocks for the formation of proteins. Among the 20 amino acids contributing to protein formation, nine cannot be synthesized either at all or in sufficient amounts by the human body, so they can only be acquired by dietary intake (i.e., they are essential AA), which include: valine, leucine, isoleucine, threonine, lysine, histidine, phenylalanine, tryptophan and methionine. Other AAs can be endogenously synthesized [13]. Circulating branched-chain AA [14], glutamine and glutamate [15,16], and AA in the tryptophan-kynurenine pathway [17] have been previously associated with the risk of atherosclerotic cardiovascular events. Consequently, it is likely that circulating levels of AA may be related to the risk of PAD. Studying the association between plasma AA and future PAD may help identify the potential role of these metabolites in the pathogenesis of PAD. Some of these AAs may contribute to the design of combined biomarkers for the early prediction of PAD.

Using a case-control study nested within the PREDIMED trial, the aim of this study was to assess the prospective association of circulating AA or their metabolites with the risk of developing PAD. A secondary aim was to examine whether these associations might be modified by a dietary intervention based on the Mediterranean diet.

## 2. Results

Table 1 shows the baseline characteristics of the participants according to PAD incidence. PAD cases were more likely to have type 2 diabetes (T2D) and belong to the control group of the trial.

The first step in the present case-control study was to analyze the association between baseline AA and PAD incidence in conditional logistic models. In the crude models, we observed that tryptophan (OR_for 1 SD increase_ = 0.80 (0.64–0.99)), serine (OR_for 1 SD increase_ = 0.72 (0.57–0.90)) and threonine (OR_for 1 SD increase_ = 0.74 (0.59–0.91)) were inversely associated with PAD risk.

When the models were further adjusted for the intervention group, smoking status, BMI, leisure-time physical activity, prevalent hypertension, prevalent T2D, prevalent dyslipidemia and educational level (Table 2), tryptophan maintained its inverse association with PAD (OR_for 1 SD increase_ = 0.78 (0.61–0.99)) although no consistent association across increasing quartiles was observed (*p*-Value for Linear trend = 0.216). Moreover, both serine and threonine were inversely associated with PAD (OR_for 1 SD increase_ = 0.67 (0.51–0.86) and 0.75 (0.59–0.95), respectively). These inverse associations were also observed across increasing quartiles of serine (*p*-Value for Linear trend = 0.015) and threonine (*p*-Value for Linear trend = 0.026). Finally, an inverse association between baseline glutamate (quartiles) and PAD risk (*p*-Value for Linear trend = 0.029) was observed, but when tested as a continuous variable (per each SD increase), the association of glutamate was only marginally significant (OR_for 1 SD increase_ = 0.79 (0.60–1.03)).

After adjusting for multiple comparisons (FDR), only the association between serine and PAD was borderline significant (*p*-value = 0.065).

The second step was to analyze the additional metabolites involved in the degradation pathways for AA that appeared to be associated with PAD development. First, the metabolites implicated in the tryptophan/kynurenine and tryptophan/serine pathways were analyzed. In Table 3, it was shown that higher values for the kynurenine to tryptophan ratio (Kyn/Trp) were associated with a higher risk of PAD. This association was observed across increasing quartiles of the Kyn/Trp ratio (*p*-Value for Linear trend = 0.012) and also when the ratio was introduced as a continuous variable (OR_for 1 SD increase_ = 1.50 (1.14–1.98)). On the other hand, baseline serotonin was associated with a higher risk of PAD across increasing quartiles (OR_Q4vsQ1_ = 2.30 (1.12–4.73); *p*-Value for Linear trend= 0.049), but this association was considerably weaker and non-significant when serotonin was introduced as a continuous variable (OR_for 1 SD increase_ = 1.18 (0.91–1.53)) (Table 3).

No metabolite related to the degradation pathways of serine and threonine was significantly associated with the risk of PAD (Table 4). The glutamine-to-glutamate ratio was not significantly associated with PAD (Table 5).

In addition, potential interactions between baseline levels of metabolites that appeared to be associated with PAD and the intervention group were analyzed. The interaction terms of the Kyn/Trp ratio and serine with the intervention group were not statistically significant (*p*-value = 0.553 (Appendix A) and *p*-value = 0.651 (Appendix A), respectively). However, an effect modification by the intervention for the association between threonine and PAD (*p*-value = 0.018) was found. The subjects allocated to either the control or the MedDiet+nuts groups and with high baseline levels of threonine showed a lower risk of PAD compared with those in the control group and with low baseline threonine levels (Figure 1). However, compared with those of low threonine levels allocated to the control group, subjects allocated to the MedDiet+EVOO group showed a lower risk of PAD regardless of their baseline threonine (Figure 1). Consequently, the beneficial effects of MedDiet+EVOO did not depend on threonine levels, while in the MedDiet+nuts group, threonine levels played a role in the association between the intervention and PAD.

## 3. Discussion

In this case-control study nested in the PREDIMED study, certain baseline AAs were prospectively associated with the risk of PAD. High circulating baseline levels of tryptophan, glutamate, serine and threonine were inversely associated with the risk of PAD. On the contrary, the ratio of kynurenine/tryptophan was prospectively associated with a higher risk of PAD.

Moreover, we found that subjects allocated to the MedDiet+EVOO group were protected against PAD regardless of their baseline AA profile. However, only subjects allocated to the control or to the MedDiet+nuts groups and with high baseline levels of threonine showed a lower risk of PAD than those with low levels of threonine.

Metabolites of the tryptophan–kynurenine degradation pathway and, specifically, the Kyn/Trp ratio have been previously associated with a higher risk of atherosclerosis [18] and coronary artery disease [19], aside from other CVDs. Indeed, in previous assessments in the PREDIMED trial, we found that the plasma Kyn/Trp ratio was associated with an increased risk of heart failure [20], and some kynurenine-related metabolites were associated with a higher risk of CVD [21]. The kynurenine pathway is intimately related to inflammation; Cytokines regulate the tryptophan–kynurenine pathway at several steps [22], and metabolites implicated in this pathway have a regulatory role in inflammatory responses [22]. In this context, inflammation is one of the processes which, in combination with endothelial dysfunction, interact with conventional risk factors leading to the onset and clinical manifestations of atherosclerosis and PAD [23,24]. Thus, it is coherent that both inflammation and the kynurenines pathway could be activated in PAD subjects even before its diagnosis or clinical manifestations. However, we cannot establish whether an activated tryptophan–kynurenine pathway may trigger the inflammatory response or whether the over-expressed cytokines activated the kynurenine pathway.

On the other hand, we found that baseline plasma serine was inversely associated with the risk of PAD. In concordance with our results, serine has been implicated in the production of antioxidants and nitric oxide in animal/cell models [25,26], which may lead to anti-atherogenic effects [25]. Serine was inversely associated with hypertension, a recognized risk factor for PAD [23] in the EPIC-Potsdam study and others supporting these findings [27].

We also found that higher circulating levels of baseline threonine were prospectively and inversely related to PAD. Similarly, threonine was previously reported to be inversely associated with some atherogenic lipids (small dense, low-density lipoprotein cholesterol, remnant-like particle cholesterol and triglycerides (TG)) in a healthy, community-based Chinese cohort [28]. In the Heart Study Framingham cohort, threonine was associated with lower levels of plasma TG [29]. Thus, the association between threonine and PAD risk might be at least partially explained by its anti-atherogenic properties.

The inverse association between MedDiet+EVOO and PAD independent of baseline levels of threonine is consistent with previous reports in the PREDIMED trial. In this trial, a MedDiet enriched with EVOO protected against PAD [30]. Surprisingly, we observed that subjects allocated to the MedDiet+nuts or the control group benefited from the intervention depending on threonine baseline levels; only those with high baseline threonine (either belonging to the MedDiet+nuts or the control group) showed a lower PAD risk as compared with control group subjects with low levels of threonine. We hypothesize that the anti-atherogenic effects of a low-fat diet or the MedDiet+nuts could act in synergy with the anti-atherogenic effects of threonine.

Finally, a potential inverse and non-linear association was observed between glutamate and PAD, which contrasts with previous findings that associated glutamate with a higher risk of atheroma plaque or CVD [16,31]. However, we did not observe any associations with glutamine or the ratio of glutamine to glutamate with PAD, which did not support the glutamate finding.

Our study presents some limitations that need to be addressed. First of all, the sample size was not large regarding the number of adjudicated incident PAD cases; thus, the results should be interpreted with caution and replicated in larger studies. Unfortunately, we were not able to replicate our results in other cohorts. Nevertheless, the results on serine and threonine (sharing metabolic pathways) seemed to be robust. On the other hand, although we have adjusted our statistical models for several confounders, residual confounding may still be present. Finally, it is not possible to generalize our results to other populations with a lower prevalence of cardiovascular risk factors, younger populations or populations pertaining to ethnic groups other than Caucasians.

In conclusion, the results of our matched case-control study nested in the PREDIMED study suggest that baseline circulating AAs, especially serine, threonine, tryptophan and the ratio kynurenine/tryptophan, may contribute to the prospective prediction of the risk of PAD in populations with high cardiovascular risk.

## 4. Materials and Methods

### 4.1. Study Design

The design and methods of the PREDIMED trial have been previously described [32]. From 2003 to 2009, 7447 Spanish participants with high cardiovascular risk were recruited in eleven different sites. Participants were female or male aged 55–80 years, with type 2 diabetes (T2D) or with at least three of the following cardiovascular risk factors: body mass index (BMI) ≥ 25 kg/m^2^, currently smoking, hypertension, high levels of LDL cholesterol, low levels of HDL cholesterol or a family history of early coronary artery disease. Participants were randomly allocated 1:1:1 to one of the three intervention groups: Mediterranean diet (MedDiet) supplemented with extra virgin olive oil (EVOO), MedDiet supplemented with mixed tree nuts or advice to adhere to a low-fat control diet. The median follow-up was 4.8 years. The study was conducted according to the guidelines of the Declaration of Helsinki and approved by the Institutional Review Board of all PREDIMED recruiting centers.

For the molecular phenotyping studies, a matched case-control study (a retrospective study designed after the completion of the main trial) was designed, including all the PAD incident cases identified from 2003 to 2017 with available blood samples. Incidence density sampling with replacement was used as the control sampling method [20]. Thus, controls were randomly selected from all participants at risk at the time of the occurrence of the incident case, and matched controls could be selected again as a control for another index case and they could become later a case [33]. One to three controls per case were matched by their recruitment center, year of birth (± five years) and sex. Initially, 170 cases and 254 controls were included in the analyses. However, there were four controls and three cases that were excluded because of missing data for the studied metabolites. Therefore, the final analyzed sample consisted of 167 PAD cases and 250 controls.

### 4.2. Ascertainment of PAD

PAD was pre-specified as a secondary endpoint in the PREDIMED trial protocol. Participants’ medical records were systematically and periodically examined during the follow-up period to identify PAD cases [30]. The adjudication of a PAD case by the Clinical Events Committee was required to meet at least one of the following documented criteria: an ankle-brachial index lower than 0.9 at rest, clinical evidence of arterial occlusive disease or an endovascular or open surgical revascularization (or amputation) occurrence. All PAD and other major cardiovascular disease-related (CVD) events (myocardial infarction, stroke and CVD death) were confirmed by the Central Endpoint Adjudication Committee, who were blinded to the interventions.

### 4.3. Sample Collection and Metabolomic Analysis

At baseline, participants provided fasting blood samples which were processed at each recruiting center no later than 2 h after collection and stored at −80 °C until analysis. Samples for this study were shipped and assayed in the same analytical run and randomly sorted to reduce bias and inter-assay variability.

Liquid chromatography-tandem mass spectrometry (LC-MS) was used to measure polar plasma metabolites. All AA were measured using hydrophilic interaction liquid chromatography (HILIC) coupled with high-resolution positive ion mode MS detection (18 out of the 20 AA were available among the annotated metabolomics data). For the specific analyses of the tryptophan pathway, kynurenine and quinolinic acid were measured using HILIC and targeted negative ion mode MS detection. Both methods were previously described [34,35].

### 4.4. Covariates

Baseline questionnaires were used to collect sociodemographic lifestyle variables, prevalent and family history of disease and medication use. Leisure-time physical activity was measured with the validated version of the Minnesota Leisure-Time Physical Activity questionnaire [36].

### 4.5. Statistical Analysis

Individual metabolite values were normalized and scaled in multiples of 1 SD with Blom’s inverse normal transformation [37]. Means and standard deviations (SD) were used to describe quantitative traits, and percentages were used to describe categorical variables.

Conditional logistic regression models were fitted to account for the matching between cases and controls. In these conditional logistic models, matched odds ratios (OR) and their 95% CIs for PAD were calculated while considering the first quartile as the reference category. Quartile cut-off points were calculated according to the distribution of metabolites among controls (participants without PAD through the follow-up). The matched odds ratios for the SD of each metabolite, including them as continuous variables, were also calculated. Crude and multivariable models adjusted for the intervention group (MedDiet + EVOO/MedDiet + nuts/control), smoking status (never/current/former), BMI (kg/m^2^), leisure-time physical activity (metabolic equivalent task [MET]-min/day), prevalent chronic conditions at baseline hypertension, T2D and dyslipidemia- and educational level (primary school or lower/secondary school or higher) were fitted. The *p*-values were penalized for multiple comparisons using the false discovering rate described by Simes [38].

For AAs that were prospectively associated with PAD incidence in the individual metabolite analyses explained above, additional analyses, including all the annotated metabolites available for each related pathway, were performed. Specifically, conditional logistic regression models adjusted for the same confounders for the available metabolites implicated in the degradation pathways of tryptophan (kynurenine, kynurenic acid, 3-hydroxyanthranilic acid and quinolinic acid), serine, threonine and glycine (choline, betaine and dimethylglycine) were performed. Ratios between implicated metabolites (kynurenine/tryptophan, serine/tryptophan and glutamine/glutamate) were also calculated to evaluate specific steps of each pathway and included in the same conditional logistic regression models.

Interactions (2 degrees of freedom) between each plasma AA as a dichotomous trait (below/above the median) and considering the three arms of the intervention (control/MedDiet + EVOO/MedDiet + nuts) were tested using conditional logistic models adjusted for smoking status (never/current/former), BMI (kg/m^2^), leisure-time physical activity (metabolic equivalent task [MET]-min/day), prevalent chronic diseases such as hypertension, T2D and dyslipidemia–, family history of premature coronary heart disease (CHD) and educational level (primary school or lower/secondary school or higher). In addition, adjustment for propensity scores that used 30 baseline variables to estimate the probability of assignment to each of the intervention groups and robust variance estimators were used considering that a small percentage of participants were non-individually randomized to the intervention groups and minor imbalances in baseline covariates existed in the trial [32]. A detailed description of how propensity scores were calculated can be found in the supplement of the main paper of PREDIMED [32]. The *p*-values for interactions were calculated using the likelihood ratio test for each assessment of potential effect modifiers.

Moreover, as an ancillary analysis, if the interaction of any of the metabolites and intervention results were statistically significant, a new logistic conditional model including a joint variable that combined the metabolite as dichotomous (below/above the median) and the intervention group (MedDiet+EVOO and MedDiet+nuts vs. control) was conducted.

A *p*-value < 0.05 was deemed statistically significant for each performed test.

All the statistical procedures were carried out with STATA 16 software, 2.1.

## Figures and Tables

**Figure 1 ijms-24-00270-f001:**
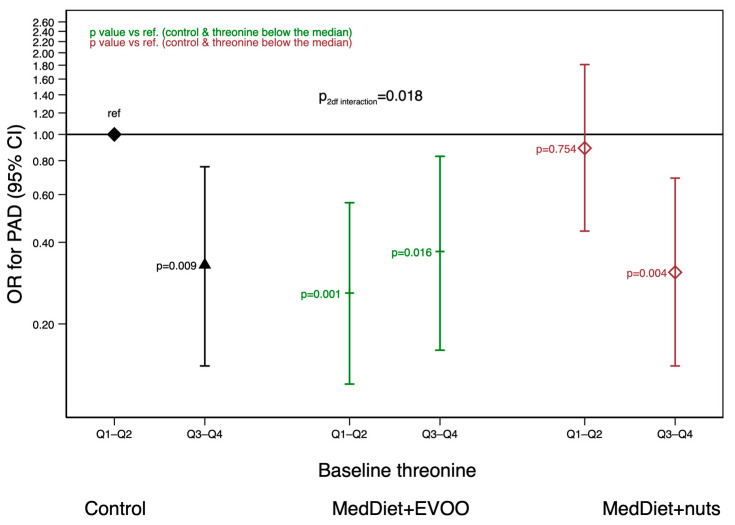
Matched odds ratios (95% CI) for the association between baseline threonine (below/above the median) and incident peripheral artery disease (PAD) within each PREDIMED intervention arm. Multivariable model adjusted for the intervention group (Mediterranean diet vs. control group), body mass index (kg/m^2^), smoking (never, current, former), leisure-time physical activity (metabolic equivalent tasks in min/d), prevalent chronic conditions at baseline (dyslipidemia, hypertension and diabetes), educational level (primary or lower/secondary or higher) and propensity scores predicting randomization to account for small between-group imbalances at baseline. MedDiet: Mediterranean diet.

**Table 1 ijms-24-00270-t001:** Baseline characteristics of the participants according to the incidence of peripheral artery disease (PAD).

	Controls (Non-Cases) (n = 250)	PAD Cases (n = 167)
Age (years)	68.0 (6.8)	67.6 (6.8)
Body mass index (kg/m^2^)	29.2 (3.3)	29.2 (3.6)
Leisure time physical activity (METs-min/d)	285.0 (288)	223.0 (221)
Female (%)	29.2	31.1
Hypertension (%)	79.2	77.2
Dyslipidemia (%)	65.2	61.1
Type 2 diabetes (%)	52.4	65.3
Smoking		
Never (%)	43.2	35.9
Former (%)	20.8	28.7
Current (%)	36	35.3
Intervention group		
Control (low-fat) (%)	30.8	38.9
MedDiet+EVOO (%)	39.6	27.5
MedDiet+nuts (%)	29.6	33.5
Education		
Primary or less (%)	73.2	71.9
Secondary (%)	17.6	22.8
University graduate (%)	9.2	5.39

Mean (standard deviation) is presented unless percentage (%) is specified.

**Table 2 ijms-24-00270-t002:** Association * between plasma amino acids and incident peripheral artery disease (PAD).

	Q1	Q2	Q3	Q4	*p*-Value for Linear Trend	Per 1 SD	FDR ** Corrected *p*-Values (Per 1 SD)
Glycine	Ref.	0.78 (0.42–1.45)	0.92 (0.49–1.71)	0.92 (0.48–1.76)	0.866	0.93 (0.73–1.18)	0.871
Alanine	Ref.	0.70 (0.37–1.35)	0.96 (0.49–1.88)	0.97 (0.51–1.85)	0.840	1.02 (0.81–1.29)	0.958
Valine	Ref.	0.77 (0.40–1.48)	0.50 (0.25–0.99)	1.01 (0.54–1.87)	0.948	1.00 (0.79–1.26)	0.991
Leucine	Ref.	1.62 (0.84–3.12)	1.28 (0.63–2.61)	1.10 (0.56–2.17)	0.918	1.04 (0.82–1.32)	0.945
Isoleucine	Ref.	1.10 (0.57–2.12)	1.29 (0.66–2.53)	0.84 (0.42–1.70)	0.627	0.99 (0.79–1.28)	0.991
Phenylalanine	Ref.	1.33 (0.68–2.59)	1.44 (0.75–2.79)	1.76 (0.90–3.44)	0.099	1.09 (0.87–1.36)	0.871
Tryptophan	Ref.	1.12 (0.60–2.11)	0.65 (0.33–1.29)	0.76 (0.38–1.52)	0.264	0.78 (0.61–0.99)	0.315
Methionine	Ref.	0.59 (0.30–1.14)	0.60 (0.31–1.16)	0.85 (0.45–1.61)	0.709	0.93 (0.73–1.19)	0.871
Proline	Ref.	1.48 (0.76–2.89)	1.28 (0.65–2.54)	1.54 (0.78–3.02)	0.286	1.17 (0.92–1.49)	0.697
Serine	Ref.	0.84 (0.46–1.54)	0.55 (0.29–1.02)	0.41 (0.20–0.84)	0.015	0.68 (0.53–0.88)	0.065
Threonine	Ref.	0.77 (0.42–1.42)	0.41 (0.20–0.84)	0.56 (0.29–1.07)	0.026	0.76 (0.60–0.97)	0.264
Tyrosine	Ref.	0.90 (0.49–1.64)	0.68 (0.35–1.32)	1.19 (0.64–2.23)	0.792	1.02 (0.81–1.29)	0.958
Asparagine	Ref.	0.46 (0.23–0.93)	0.72 (0.37–1.38)	0.77 (0.41–1.44)	0.795	0.91 (0.72–1.16)	0.871
Glutamine	Ref.	0.65 (0.34–1.24)	0.70 (0.35–1.39)	0.81 (0.42–1.55)	0.556	0.93 (0.72–1.18)	0.871
Glutamate	Ref.	0.41 (0.21–0.79)	0.62 (0.34–1.27)	0.42 (0.20–0.88)	0.029	0.78 (0.59–1.03)	0.381
Histidine	Ref.	0.74 (0.38–1.45)	0.66 (0.34–1.15)	1.16 (0.61–2.19)	0.684	1.10 (0.87–1.40)	0.871
Lysine	Ref.	0.68 (0.36–1.26)	0.84 (0.45–1.55)	0.81 (0.44–1.50)	0.625	0.95 (0.76–1.20)	0.945
**Arginine**	Ref.	1.52 (0.82–2.81)	1.22 (0.66–2.26)	1.45 (0.74–2.83)	0.387	1.11 (0.88–1.40)	0.871

* Multivariable model adjusted for intervention group, body mass index, smoking, leisure-time physical activity, educational level, dyslipidemia, hypertension and diabetes.** FDR: false discovery rate.

**Table 3 ijms-24-00270-t003:** Association * between plasma tryptophan–kynurenine/tryptophan–serotonin pathway metabolites and incident peripheral artery disease (PAD).

	Q1	Q2	Q3	Q4	*p*-Value for Linear Trend	Per 1 SD
Tryptophan	Ref.	1.12 (0.60–2.11)	0.65 (0.33–1.29)	0.76 (0.38–1.52)	0.264	0.78 (0.61–0.99)
Kynurenine	Ref.	1.16 (0.61–2.24)	1.30 (0.65–2.59)	1.74 (0.87–3.47)	0.177	1.23 (0.94–1.59)
Ratio Kyn/Trp **	Ref.	1.75 (0.87–3.52)	1.72 (0.84–3.50)	3.11 (1.42–6.82)	0.012	1.50 (1.14–1.98)
Kynurenic acid	Ref.	0.85 (0.44–1.65)	0.83 (0.44–1.58)	1.50 (0.80–2.83)	0.284	1.20 (0.96–1.51)
Hydroxyanthranilic acid	Ref.	0.64 (0.33–1.23)	1.27 (0.67–2.40)	0.82 (0.38–1.77)	0.779	0.92 (0.69–1.22)
Quinolinic acid	Ref.	0.76 (0.38–1.50)	0.69 (0.35–1.35)	1.08 (0.52–2.22)	0.988	0.90 (0.69–1.18)
Serotonin	Ref.	1.56 (0.77–3.14)	1.35 (0.70–2.60)	2.30 (1.12–4.73)	0.049	1.18 (0.91–1.53)
Serotonin/Trp ratio	Ref.	1.53 (0.77–3.02)	1.24 (0.62–2.48)	1.82 (0.92–3.60)	0.140	1.22 (0.95–1.57)

* Multivariable model adjusted for the intervention group, body mass index, smoking, leisure-time physical activity, education level, dyslipidemia, hypertension and diabetes. ** Kynurenine/tryptophan ratio.

**Table 4 ijms-24-00270-t004:** Association * between plasma serine–glycine–threonine pathway metabolites and incident of peripheral artery disease (PAD).

	Q1	Q2	Q3	Q4	*p*-Value for Linear Trend	Per 1 SD
Serine	Ref.	0.84 (0.46–1.54)	0.55 (0.29–1.02)	0.41 (0.20–0.84)	0.015	0.67 (0.51–0.86)
Glycine	Ref.	0.78 (0.42–1.45)	0.92 (0.49–1.71)	0.92 (0.48–1.76)	0.866	0.93 (0.73–1.18)
Threonine	Ref.	0.77 (0.42–1.42)	0.41 (0.20–0.84)	0.56 (0.29–1.07)	0.026	0.75 (0.59–0.95)
Choline	Ref.	0.89 (0.47–1.68)	0.69 (0.36–1.32)	1.14 (0.60–2.19)	0.823	1.02 (0.80–1.29)
Betaine	Ref.	1.04 (0.57–1.91)	0.92 (0.49–1.76)	0.95 (0.45–2.04)	0.821	1.03 (0.80–1.32)
Dimethylglycine	Ref.	0.74 (0.38–1.42)	0.93 (0.48–1.78)	0.77 (0.39–1.53)	0.552	0.96 (0.76–1.21)

* Multivariable model adjusted for the intervention group, body mass index, smoking, leisure-time physical activity, education level, dyslipidemia, hypertension and diabetes.

**Table 5 ijms-24-00270-t005:** Association between plasma glutamine–glutamate pathway metabolites and incident of peripheral artery disease (PAD). Multivariable model adjusted for the intervention group, body mass index, smoking, leisure-time physical activity, education level, dyslipidemia, hypertension and diabetes.

	Q1	Q2	Q3	Q4	*p*-Value for Linear Trend	Per 1 SD
Glutamine	Ref.	0.65 (0.34–1.24)	0.70 (0.35–1.39)	0.81 (0.42–1.55)	0.556	0.93 (0.73–1.18)
Glutamate	Ref.	0.41 (0.21–0.79)	0.62 (0.34–1.15)	0.42 (0.20–0.88)	0.029	0.78 (0.59–1.03)
Gln/Glt ratio *	Ref.	1.35 (0.68–2.69)	1.21 (0.57–2.56)	2.03 (0.99–4.15)	0.059	1.22 (0.92–1.60)

* Multivariable model adjusted for the intervention group, body mass index, smoking, leisure-time physical activity, education level, dyslipidemia, hypertension and diabetes.

## Data Availability

We will be happy to provide access to the Predimed dataset (including data dictionaries), making possible the replication of the main analyses used for the present article. Due to the restrictions imposed by the Informed Consent and the Institutional Review Board, bona fide investigators interested in analyzing the Predimed dataset used for the present article may submit a brief proposal and statistical analysis plan to the corresponding author. Upon approval from the Predimed Steering Committee and Institutional Review Boards, the data will be made available to them using an onsite secure access data enclave.

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
