# Peer review of "Circulating Amino Acids and Risk of Peripheral Artery Disease in the PREDIMED Trial"

_ijms, 2022, doi:10.3390/ijms24010270_

Round 1

Reviewer 1 Report

In this study, the authors aimed to assess the prospective association of circulating AA or their metabolites with the risk of developing PAD. Our secondary aim was to examine whether these associations might be modified by a dietary intervention based on the Mediterranean diet.  The study was very well designed and the statistical methods were also rigorous. There were some suggestions for the authors.

[ Methods ]

1. For sample collection and metabolomic analysis, all participants have tested at baseline (sample 1) , and how about tests (sample 2) after intervention ?

[Result ]

1. In the 4th paragraph of result, "On the other hand, baseline serotonin was associated with a higher risk of PAD across increasing quartiles (ORQ4vsQ1=2.30 (1.12-4.73); Pfor linear trend=0.049) but this association was considerably weaker and non-significant when serotonine was introduced as a continuous variable (ORfor 1 SD increase=1.18 (0.91-1.53))" . The Reference should be table 3 instead of table 4.

2. In the 5th paragraph of the "result rection",   "We did not find any metabolite related to the degradation  pathways of serine and threonine significantly associated with the  risk of PAD. The glutamine to glutamate ratio was neither significantly associated with PAD ". The Reference should be table 4 instead of table 5. 

3. In the 6th paragraph of the "result rection",  this paragraph "* Multivariable model adjusted for intervention group, body

mass index, smoking, leisure-time physical activity, education

level, dyslipidemia, hypertension and diabetes.  "didn't link to the context clues.   I thinik this paragraph should describe what the author found in table 5.

4. In the 7th paragraph of the "result rection",  the authors found an effect modification by intervention for the association between threonine and PAD (p=0.018). Thus, finally, did the authors explained which (threonine or intervention)  have more impact on the risk of PAD ?  or can we interpretate that intervention "MedDiet+ EVOO or MedDiet+nuts" have beneficial effect on PAD only in the high threonine condition?

5. From Figure 1, to compare the control group and the MedDiet+nuts group, we can see the risk (OR:  of Q1-2 low threonine level) was not lower in the MedDiet+nuts group than that in the control (low fat diet group). Does that mean that "MedDiet+nuts" didn't have beneficial effect on PAD in low threonine level.

[Discussion]

1. Can the authors explain how Diet intervention have interaction with certain amino acid levels, and thus alter outcome risks ?

2. The author suggest that certain amino acid level can serve as biomarker for PAD. Does the author mean that these amino acid level can be considered as prognostic biomarkers ( in spite of intervention or not) ; or that these amino acid level can be considered as predictive biomarkers ( thus we need amino acid levels before and after intervention) ?

3. Are these baseline circulating AA, especially serine, threonine, tryptophan can be modified by intervention?

4. For intervention, can you summarize your suggestion for the high risk CVD patients in the last paragraph of discussion ? What evidence do we have to suggest these patients to received MedDiet+EVOO ?

[ Tables and figures ] 

1. In Table 1 , there are two "control: row 1 and row 14" , which is confusing. Please consider using the terms  : (1) high risk CAD "controls" in the first row of the table ; (2) for the intervention subset, low-fat diet controls.

2. For Table 2, please also explained what level is considered as statistically significant for 'p-for linear trend ' in the legend ; or  annotate in the last part of the  " Statistical analysis " section.

Author Response

Reviewer 1

In this study, the authors aimed to assess the prospective association of circulating AA or their metabolites with the risk of developing PAD. A secondary aim was to examine whether these associations might be modified by a dietary intervention based on the Mediterranean diet.  The study was very well designed and the statistical methods were also rigorous. There were some suggestions for the authors.

Thank you for your deep revision and your kind comments. We have revised the manuscript to improve the writing (English) following your suggestion.

Methods

  1. For sample collection and metabolomic analysis, all participants have tested at baseline (sample 1), and how about tests (sample 2) after intervention?

For the present study, a nested matched case-control design was used and only the baseline metabolome was determined. The effect of the intervention in metabolome changes was not analyzed in this particular case-control study. Nevertheless, it was already reported in other previous studies (please check: Li J, et al. Eur Heart J 2020, PMID: 32406924, and similar sources). Furthermore, the specific metabolomic signature of each arm of the intervention after one year will be the aim of a future specific study on which we are working now.

[Results]

  1. In the 4th paragraph of result, "On the other hand, baseline serotonin was associated with a higher risk of PAD across increasing quartiles (ORQ4vsQ1=2.30 (1.12-4.73); Pfor linear trend=0.049) but this association was considerably weaker and non-significant when serotonine was introduced as a continuous variable (ORfor 1 SD increase=1.18 (0.91-1.53))" . The Reference should be table 3 instead of table 4.

Corrected. We apologize for this typo.

  1. In the 5th paragraph of the "result rection", "We did not find any metabolite related to the degradation pathways of serine and threonine significantly associated with the risk of PAD. The glutamine to glutamate ratio was neither significantly associated with PAD ". The Reference should be table 4 instead of table 5.

Corrected. Sorry for this typo. 

  1. In the 6th paragraph of the "result rection", this paragraph "* Multivariable model adjusted for intervention group, body mass index, smoking, leisure-time physical activity, education level, dyslipidemia, hypertension and diabetes. "didn't link to the context clues. I thinik this paragraph should describe what the author found in table 5.

This was a mismatch. It seems a formatting error of the journal. In fact, the sentence "* Multivariable model adjusted for intervention group, body mass index, smoking, leisure-time physical activity, educational level, dyslipidemia, hypertension and diabetes.” is the legend of table 2, so it seems that it is duplicated by some issue related to the formatting of the tables in the processing of the manuscript.

  1. In the 7th paragraph of the "result rection", the authors found an effect modification by intervention for the association between threonine and PAD (p=0.018). Thus, finally, did the authors explained which (threonine or intervention) have more impact on the risk of PAD ? or can we interpretate that intervention "MedDiet+ EVOO or MedDiet+nuts" have beneficial effect on PAD only in the high threonine condition?

We now have tried to better explain this finding. We observed different effects for threonine depending on the study group. In the case of the MedDiet+EVOO group, the beneficial effects of the intervention (protection against PAD) was observed independently of the levels of threonine. However, for the control and MedDiet+Nuts groups, we observed that high levels of threonine were associated with a lower risk of PAD.

We have added a new sentence at the end of the 7th paragraph to better clarify this finding:

“Consequently, the beneficial effects of MedDiet+EVOO did not depend on threonine levels while in the MedDiet+Nuts group threonine levels played a role in the association between the intervention and PAD.”

  1. From Figure 1, to compare the control group and the MedDiet+nuts group, we can see the risk (OR: of Q1-2 low threonine level) was not lower in the MedDiet+nuts group than that in the control (low fat diet group). Does that mean that "MedDiet+nuts" didn't have beneficial effect on PAD in low threonine level.

You are right, the higher the threonine the lower the risk of PAD in the MedDiet+Nuts group. However, no effect was found for MedDiet+Nuts & low levels of threonine.

[Discussion]

  1. Can the authors explain how Diet intervention have interaction with certain amino acid levels, and thus alter outcome risks?

Thank you for raising this issue. Due to the nature of our study (case-control design with only baseline measurements) we cannot establish if the intervention affected the levels of these AA but it is possible that the properties of the MedDiet may counteract or boost the effects of certain AA on PAD. Specifically, in the case of threonine, which showed a statistically significant interaction with the intervention, we hypothesized that it is possible that the anti-atherogenic effects of the intervention with the MedDiet+nuts could act in synergy with the antiatherogenic effects of baseline threonine. Thus, the intervention boosted the anti-atherogenic effects of high baseline threonine levels.

  1. The author suggest that certain amino acid level can serve as biomarker for PAD. Does the author mean that these amino acid level can be considered as prognostic biomarkers (in spite of intervention or not) ; or that these amino acid level can be considered as predictive biomarkers ( thus we need amino acid levels before and after intervention) ?

Our study suggests that serine, threonine, tryptophan, and the ratio kynurenine/tryptophan may contribute (in combination with traditional risk factors) to predict the risk of PAD, independently of the intervention. Specifically, for a high cardiovascular risk population.

  1. Are these baseline circulating AA, especially serine, threonine, tryptophan can be modified by intervention?

We cannot know in this cross-sectional study. However, in a previous study in PREDIMED it was shown that the intervention may be potentially modifying tryptophan and its related metabolites.

Yu E, Papandreou C, Ruiz-Canela M, Guasch-Ferre M, Clish CB, Dennis C, Liang L, Corella D, Fitó M, Razquin C, Lapetra J, Estruch R, Ros E, Cofán M, Arós F, Toledo E, Serra-Majem L, Sorlí JV, Hu FB, Martinez-Gonzalez MA, Salas-Salvado J. Association of Tryptophan Metabolites with Incident Type 2 Diabetes in the PREDIMED Trial: A Case-Cohort Study. Clin Chem. 2018;64:1211-1220. doi: 10.1373/clinchem.2018.288720

  1. For intervention, can you summarize your suggestion for the high risk CVD patients in the last paragraph of discussion ? What evidence do we have to suggest these patients to received MedDiet+EVOO ?

It was already published in the main paper analyzing the effect of intervention on PAD in the PREDIMED trial, that both MedDiets, and especially, MedDiet+EVOO were associated with a lower risk of PAD.

Ruiz-Canela, M.; Estruch, R.; Corella, D.; Salas-Salvadó, J.; Martínez-González, M.A. Association of Mediterranean Diet with Peripheral Artery Disease: The PREDIMED Randomized Trial. JAMA 2014, 311, 415–417, doi:10.1001/JAMA.2013.280618.

Thus, it was not an objective of this study to analyze again the effect of intervention on PAD. We have clarified in the discussion that this finding related to the effect of intervention on PAD was already reported in PREDIMED.

[ Tables and figures ] 

  1. In Table 1 , there are two "control: row 1 and row 14" , which is confusing. Please consider using the terms: (1) high risk CAD "controls" in the first row of the table ; (2) for the intervention subset, low-fat diet controls.

Thank you for your comment. We have clarified this issue in table 1.

Controls (non-cases)

(n=250)

PAD cases

(n=167)

Age

68.0 (6.8)

67.6 (6.8)

BMI

29.2 (3.3)

29.2 (3.6)

Leisure time physical activity (METs-min/d)

285.0 (288)

223.0 (221)

Female (%)

29.2

31.1

Hypertension (%)

79.2

77.2

Dyslipidaemia (%)

65.2

61.1

Type 2 diabetes (%)

52.4

65.3

Smoking

  Never (%)

43.2

35.9

  Former (%)

20.8

28.7

  Current (%)

36

35.3

Intervention group

  Control (Low-fat) (%)

30.8

38.9

  MedDiet+EVOO (%)

39.6

27.5

  MedDiet+ Nuts (%)

29.6

33.5

Education

  Primary or less (%)

73.2

71.9

  Secondary (%)

17.6

22.8

  University graduate (%)

9.2

5.39

  1. For Table 2, please also explained what level is considered as statistically significant for 'p-for linear trend ' in the legend ; or annotate in the last part of the " Statistical analysis " section.

We have added this information at the end of the methods section (statistical analyses):

P-value<0.05 was considered statistically significant for each performed test.”

Reviewer 2 Report

This study aimed to prospectively analyze the associations of baseline levels of plasma amino acids with subsequent risk of peripheral artery disease. The second intention was to evaluate the potential effect of modification by a nutritional intervention with Mediterranean diet (MedDiet). I have a few questions listed below:

1. Line 143-145: “One to three controls per case were matched by recruitment center, year of birth (± 5 years), and sex”. Could you please provide more explanation about how this was executed in the study? How did the research staff determine the numbers (1, 2, or 3) of controls to be selected? And how did the research staff reach the candidates for controls?

2. Line 147: Why there were missing data for the studied metabolites?

3. Line 185-242: Could you please explain how the sample size was determined in this prospective study?

4. Line 124: Please provide the description for the abbreviation “T2D”.

Author Response

Reviewer 2

This study aimed to prospectively analyze the associations of baseline levels of plasma amino acids with subsequent risk of peripheral artery disease. The second intention was to evaluate the potential effect of modification by a nutritional intervention with Mediterranean diet (MedDiet). I have a few questions listed below:

Thank you for detailed revision and your suggestions.

  1. Line 143-145: “One to three controls per case were matched by recruitment center, year of birth (± 5 years), and sex”. Could you please provide more explanation about how this was executed in the study? How did the research staff determine the numbers (1, 2, or 3) of controls to be selected? And how did the research staff reach the candidates for controls?

For each PAD case, a pool of candidates to be controls was generated considering the follow-up (free of PAD at the time of endpoint diagnosis in the matched case), and the matching variables: recruitment center, year of birth (± 5 years), and sex. Up to 3 controls (depending on the availability of samples) were randomly selected from the pool of matched and disease-free study participants. This means that, if possible (all the conditions were met and samples were available), three controls were selected for each case.

We decided to select more than one control per each case because the higher the number of controls per case the higher the statistical power.

  1. Line 147: Why there were missing data for the studied metabolites?

Because not all samples had enough quality as to provide reliable measurements and it was impossible to measure either the negative or the positive metabolites.

  1. Line 185-242: Could you please explain how the sample size was determined in this prospective study?

This study was designed in parallel with two other case-control studies, for atrial fibrillation and heart failure, nested within the PREDIMED trial:

For power analysis, the sample size for different outcomes would be 594 pairs of AF case-controls, 332 pairs of HF case-controls, 196 pairs of PAD case-controls, and 986 pairs of composite CVD case-controls. Although we will use FDR to adjust for multiple testing, we demonstrated sufficient power using an even more stringent Bonferroni correction for 400 metabolites. Please find below the simulations that we used to compute the statistical power in advance.

We computed the power to detect a metabolite associated with HF, AF, PAD and composite CVD risk with odds ratio (OR) ranges from 1.2 to 2.0 (corresponding to one standard deviation increase in metabolite level) at nominal false positive rate of 0.05 or 0.05/400 (Figure 5A for 1-1 matched design). Even at the conservative Bonferroni correction threshold of 0.05/400, we will have 80% power to detect OR for the composite CVD (1.30), HF (1.59), AF (1.41) and PAD (1.86) using 1-1 matched case-control pairs or the composite CVD (1.30), HF (1.37), AF (1.34) and PAD (1.40) using the combined controls.

(See the graph in the attached rebuttal letter)

  1. Line 124: Please provide the description for the abbreviation “T2D”.

Done

Reviewer 3 Report

The authors reported a well-conducted matched case-control study using the PREDIMED trial data, I have some comments:

1. Is this study pre-defined at the beginning of the PREDIMED trial, or retrospective analysis of the trial? I think It's better to make it clear.

2. Among the 7447 participants, how many provided the fasting blood samples? If it is possible don't to break the randomization, I would like to know, If all participants with fasting blood samples be used, without matching, such as using Cox regression, could similar results be obtained. 

3. The False Discovery Rate should be controlled. Are the pathways metabolites and the ratios of the metabolites pre-defined or post-hoc ones based on the data? I think it's important and better to make it clear.

4. For the interaction analysis, was the propensity score (which is stated in the statistical analysis part) developed in this study, if it was, more details better be provided.  And from my point of view, it's better to provide the results of all analyzed variables, not just the significant ones.  

5. In the propensity score part, it mentioned that there were 30 baseline variables, but in table 1, the variables are far less than 30, please give some explanation.

Author Response

Reviewer 3

The authors reported a well-conducted matched case-control study using the PREDIMED trial data, I have some comments:

Thank you for your deep revision and your kind comments.

  1. Is this study pre-defined at the beginning of the PREDIMED trial, or retrospective analysis of the trial? I think It's better to make it clear.

It was a retrospective study designed after the completion of the main trial. We have added this information in the methods section to comply with your suggestion.

  1. Among the 7447 participants, how many provided the fasting blood samples? If it is possible don't to break the randomization, I would like to know, If all participants with fasting blood samples be used, without matching, such as using Cox regression, could similar results be obtained. 

All the participants in PREDIMED provided biological samples. However, in the present study we didn’t use all the randomized participants, instead we adopted a nested and matched case-control study for reasons of efficiency (Rose S, Laan MJ. Why match? Investigating matched case-control study designs with causal effect estimation. Int J Biostat. 2009;5:1; PMID: 20231866). We used incidence density sampling of controls to emulate a cohort study that measures rates and therefore uses person-time denominators. In a cohort study using person-time denominators, each person contributes a varying amount of information, as measured by their person-time contribution. Random sampling of controls from that person-time experience does not result in each person having an equal chance of being selected as a control. Rather, the sampling probability of any person as a control would be proportional to the amount of person-time that person spends at risk of disease in the source population (please see Lash TL, VanderWeele TJ, Haneause S, Rothman K. Modern Epidemiology, 4th ed. Chapter 8. Wolters Kluwer Health, 2021. The matching was performed not only according to the follow-up time (incidence density sampling), but also by sex and age.

  1. The False Discovery Rate should be controlled. Are the pathways metabolites and the ratios of the metabolites pre-defined or post-hoc ones based on the data? I think it's important and better to make it clear.

Following your suggestion, we have added the FDR correction (for the analyses per 1 SD increase) in the main table of the paper (table 2). The ratios were pre-defined. In fact, previously published papers in PREDIMED included these ratios when analyzing tryptophan or glutamine pathways.

Zheng, Y.; Hu, F.B.; Ruiz-Canela, M.; Clish, C.B.; Dennis, C.; Salas-Salvado, J.; Hruby, A.; Liang, L.; Toledo, E.; Corella, D.; et al. Metabolites of Glutamate Metabolism Are Associated With Incident Cardiovascular Events in the PREDIMED PREvención Con DIeta MEDiterránea (PREDIMED) Trial. J Am Heart Assoc 2016, 5, doi:10.1161/JAHA.116.003755.

Yu E, Papandreou C, Ruiz-Canela M, Guasch-Ferre M, Clish CB, Dennis C, Liang L, Corella D, Fitó M, Razquin C, Lapetra J, Estruch R, Ros E, Cofán M, Arós F, Toledo E, Serra-Majem L, Sorlí JV, Hu FB, Martinez-Gonzalez MA, Salas-Salvado J. Association of Tryptophan Metabolites with Incident Type 2 Diabetes in the PREDIMED Trial: A Case-Cohort Study. Clin Chem. 2018;64:1211-1220. doi: 10.1373/clinchem.2018.288720

  1. For the interaction analysis, was the propensity score (which is stated in the statistical analysis part) developed in this study, if it was, more details better be provided. 

No, the propensity scores (PS) were not specifically calculated for this study. In fact, they were previously used in the main analyses of PREDIMED and in the subsequent analyses and sub-studies. The reference to the main paper of PREDIMED where the detailed description of those propensity scores can be found was already included in the previous version of the manuscript. Additionally, in the new version, we have also added that the complete information can be found in the supplementary appendix of the main paper of PREDIMED (Estruch et al. N Engl J Med 2018; PMID: 29897866). We add below the complete description of how these PS were calculated and what they are used for.

Because of minor imbalances in the randomization procedures of the whole PREDIMED trial, we analyzed our data to estimate the associations between the interventions and outcomes, using methods that do not exclusively rely on the assumption that all patients had been randomly assigned to the treatment groups (PMID: 29897866). Consequently, when we analyzed the effects of the intervention, we included as covariates propensity scores estimating the probability of assignment to each of the intervention arms of the trial. These propensity scores were estimated by using a multinomial logistic model with the allocation (3 arms) of the trial as the outcome (dependent variable, with 3 categories) and the following 30 baseline variables as predictors of the allocation (independent variables): ethnicity, marital status (3 categories), living alone, unemployment, retirement, housewife as the only occupation, presence of any disability, years of education (continuous), dyspnea, history of non-atherosclerotic cardiovascular disease, history of kidney disease, history of chronic lung disease, history of depression, cataracts, history of obstructive sleep apnea, history of cancer, use of vitamin/mineral supplements, use of angiotensin-converting enzyme inhibitors, use of diuretics, use of other anti-hypertensive medication, use of statins, use of other lipid-lowering medication, use of insulin, use of oral antidiabetic agents, use of aspirin/antiplatelet therapy, score of psychological tension (continuous, 0 to 10), fasting plasma glucose (continuous), ratio of blood total cholesterol to HDL-cholesterol (continuous), blood LDL-cholesterol levels (continuous) and blood triglycerides (continuous). After fitting this multinomial logistic model, we retained the post-estimation predicted probabilities to be allocated to each of the two active intervention diets (P1=probability of allocation to the Mediterranean diet with extra-virgin olive oil and P2=probability of allocation to the Mediterranean diet with nuts). The propensity score to be allocated to the control group (P3) is the complementary of the sum of P1+P2, i.e., the sum P1+P2+P3 should always be 1 for each participant. Therefore, P3 would be redundant and there is no need to include it in the model. In a subsequent step, we added the 2 estimated propensity scores for the intervention (P1 and P2) as continuous covariates (independent variables) in the Cox model in order to adjust for the predicted probability to be allocated to each of the two active interventions. All this information is available in the supplement of main paper of PREDIMED (PMID: 29897866). However, the rationale to use the propensity scores was to avoid any possibility of residual confounding when assessing the effect of the intervention on clinical outcomes, and in the present paper we are not assessing any effect of the intervention on clinical outcomes.

And from my point of view, it's better to provide the results of all analyzed variables, not just the significant ones.  

Following your suggestion, we have added the results related to non-significant interactions in the supplementary material. In the current version two supplementary figures are included: Fig. 1S showing the non-significant interaction intervention*Kyn/Trp and Fig. 2S representing the non-significant interaction intervention*serine and

  1. In the propensity score part, it mentioned that there were 30 baseline variables, but in table 1, the variables are far less than 30, please give some explanation.

As explained above, propensity scores were used to correct for minor imbalances in the randomization. But they were not used as covariates for the study.

Reviewer 4 Report

1. The references are up-to-date.

2. The design and research are designed in the proper way.

3. The tables and figures are clear.

4. The general context is understandable but it can be better written in the third-personal perspective.

5. The context of the "Control" is good but the formating is confusing at the first glance of the provided PDF file.  

Author Response

  1. The references are up-to-date.
  2. The design and research are designed in the proper way.
  3. The tables and figures are clear.

We are very grateful for your nice comments.

  1. The general context is understandable but it can be better written in the third-personal perspective.

According to your suggestion, we have revised the grammar to rewrite the manuscript in the third-personal perspective. We have also revised all the manuscript to improve the English as suggested by reviewer 1.

  1. The context of the "Control" is good but the formatting is confusing at the first glance of the provided PDF file.  

Thank you for your comment. We have clarified this issue in table 1.

Round 2

Reviewer 3 Report

The authors have solved my questions, I have no further comments.